# Co-Expression Networks in Sunflower: Harnessing the Power of Multi-Study Transcriptomic Public Data to Identify and Categorize Candidate Genes for Fungal Resistance

**DOI:** 10.3390/plants12152767

**Published:** 2023-07-25

**Authors:** Andrés I. Ribone, Mónica Fass, Sergio Gonzalez, Veronica Lia, Norma Paniego, Máximo Rivarola

**Affiliations:** Instituto de Agrobiotecnología y Biología Molecular (IABIMO), CICVyA—Instituto Nacional de Tecnología Agropecuaria (INTA), Consejo Nacional de Investigaciones Científicas y Técnicas (CONICET), Los Reseros y Nicolás Repetto, Hurlingham 1686, Argentina; ribone.andres@inta.gob.ar (A.I.R.); fass.monica@inta.gob.ar (M.F.); gonzalez.sergio@inta.gob.ar (S.G.); lia.veronica@inta.gob.ar (V.L.); paniego.norma@inta.gob.ar (N.P.)

**Keywords:** transcriptomics, co-expression networks, fungal pathogens, WRKY, plant pathology, meta-analysis, multi-study analysis, sunflower, candidate genes

## Abstract

Fungal plant diseases are a major threat to food security worldwide. Current efforts to identify and list *loci* involved in different biological processes are more complicated than originally thought, even when complete genome assemblies are available. Despite numerous experimental and computational efforts to characterize gene functions in plants, about ~40% of protein-coding genes in the model plant *Arabidopsis thaliana* L. are still not categorized in the Gene Ontology (GO) Biological Process (BP) annotation. In non-model organisms, such as sunflower (*Helianthus annuus* L.), the number of BP term annotations is far fewer, ~22%. In the current study, we performed gene co-expression network analysis using eight terabytes of public transcriptome datasets and expression-based functional prediction to categorize and identify *loci* involved in the response to fungal pathogens. We were able to construct a reference gene network of healthy green tissue (GreenGCN) and a gene network of healthy and stressed root tissues (RootGCN). Both networks achieved robust, high-quality scores on the metrics of guilt-by-association and selective constraints versus gene connectivity. We were able to identify eight modules enriched in defense functions, of which two out of the three modules in the RootGCN were also conserved in the GreenGCN, suggesting similar defense-related expression patterns. We identified 16 WRKY genes involved in defense related functions and 65 previously uncharacterized *loci* now linked to defense response. In addition, we identified and classified 122 *loci* previously identified within QTLs or near candidate *loci* reported in GWAS studies of disease resistance in sunflower linked to defense response. All in all, we have implemented a valuable strategy to better describe genes within specific biological processes.

## 1. Introduction

Sunflower (*Helianthus annuus* L.) is one of the most important crops for the production of high-quality oil and seeds consumed by both humans and livestock. Last year’s FAO scientific review stated that climate change represents an unprecedented challenge to food security, and due to its impact, plant diseases that ravage economically important crops are becoming more destructive and pose an increasing threat to food security and the environment [1]. A plant’s response to stress is complex, varies in time (e.g., early or delayed responses), and differs in space (e.g., whole plant, organs, tissues, or cell types). It is also governed by networks that are likely to involve numerous signaling pathways and genes [2]. Moreover, abiotic and biotic stresses can interact in complex ways. Over the past decades, many groups have conducted genetic, genomic, and transcriptomic experiments at different stages and under different conditions to understand the regulatory and genetic basis for many stress-related responses in sunflower [3,4,5,6,7]. Yet the identification of causal genes in these cases is very limited, mainly due to the lack of statistical power and poor experimental design. It is known that *Helianthus* species possess a rich spectrum of non-redundant defense mechanisms, many of which have been introduced into cultivated sunflowers from their wild relatives [8,9]. Moreover, fungal resistance in crops is typically governed by many interacting *loci*, which each offer small contributions [10]. Identifying those causal *loci* via strategies such as genome-wide association studies (GWAS) and quantitative trait *loci* mapping (QTL mapping) is very challenging and cumbersome. The genomic regions surrounding GWAS markers and those encompassing QTL mapping intervals can be extensive, including hundreds of genes, only a few of which may be potentially causal. To make matters worse, the sunflower genome is only partially annotated. Only ~60% of the genes are annotated with at least one Gene Ontology (GO) term, and all annotations are in silico (inferred from phylogeny (IBA) or electronic (IEA) annotation). Moreover, only ~22% of gene products have at least one BP GO annotation, and only 864 (1.6%) out of the total gene products are annotated as “defense response”, whereas this figure rises to 8% in *Arabidopsis thaliana* L. [11,12].

To date, several novel automated function prediction methods have been developed to address this knowledge gap. Among them, top scoring methods incorporate algorithms utilizing gene co-expression data to infer biological functions through the Guilt-By-Association principle [13,14]. Historically, BLAST [15] and InterProScan [16], using sequence similarity and the presence/absence of domains, respectively, have been the major annotation pipelines. The top scoring methods include machine learning techniques and novel ways to integrate genomic features, sequence intrinsic properties, protein-protein interactions, and other omics data and transfer these new annotations to unknown genes [17]. As expected, the assignment of molecular function (MF) GO terms significantly outperforms the assignment of BP GO terms in these machine learning methods [18]. Moreover, given the scarcity of data from different functional omics studies, such as protein interaction assays, in non-model organisms, such as sunflower, gene-expression-based automated functional prediction remains of great importance, especially for the prediction of BP GO terms [13,19]. Therefore, re-analyzing transcriptome data as a whole enhances functional annotation by leveraging the collective knowledge gained from multiple experiments. To date, several authors have focused on multi-study co-expression networks to gain a deeper understanding of specific biological processes [20], for example, in the model plant *Arabidopsis thaliana* [13,14]. The work of Depuydt et al. [13] in *A. thaliana* showed very promising results in annotating unknown genes with multi-study co-expression networks, especially using the BP GO terms under the category “response to stimulus”. Taken together, these network analyses can provide new knowledge for the field under study.

In this work, we investigated the potential of gene co-expression networks generated from multi-study datasets from all public RNAseq bio-projects currently available for sunflower to identify and/or validate candidate *loci* for defense or fungal resistance. We focused on identifying and categorizing gene candidates for two major fungal diseases, Verticillium wilt and Sclerotinia head rot. *Verticillium dahliae* is a soil-borne and seed-borne pathogen that is capable of infecting hundreds of plant species, including sunflower. Infection by *V. dahliae* causes vascular discoloration and systemic wilting [21]. On the other hand, the necrotrophic fungus *Sclerotinia sclerotiorum* can behave as an air-borne and a soil-borne pathogen and is the causal agent of three distinct diseases in sunflower (Sclerotinia root rot, Sclerotinia stem rot, and Sclerotinia head rot) [22]. Altogether, these pathogens affect oil quality and, under favorable conditions, may lead to massive production losses [23]. Different studies have identified many candidate genes potentially involved in the resistance against these diseases [21,24,25,26], but, as we mentioned, narrowing down the culprits is difficult.

Here we present the first large-scale co-expression network for sunflower and perform functional prediction to further characterize previous candidate *loci* for *S. sclerotiorum* and *V. dahliae* resistance and identify novel *loci* involved in defense response, specifically to fungal pathogens.

## 2. Results

### 2.1. Construction of a Reference Weighted Gene Co-Expression Network from Green Healthy Tissue

We generated a stress-free reference network in order to organize genes into biologically relevant clusters. We analyzed 733 sunflower samples of healthy photosynthetic tissues from 27 different studies (see Appendix A). Outlier samples were filtered out using a recursive process of read count normalization, batch effect correction, and hierarchical clustering (Figure 1, see Section 4). As a result, a total of 15,978 genes and 686 samples remained (Table 1). With this data, we calculated a co-expression network using WGCNA [27]. We selected a signed-hybrid formula with a power β = 4 as the soft threshold to ensure a scale-free network (R^2^ = 0.97). The resulting network, from now on referred to as “GreenGCN”, presented 62 modules (named Green1 to Green62) grouping 13,099 genes; the remaining 2879 genes (18%) were not included in any gene cluster and were listed in the module Green0. The network presented 198 “hub-genes”, i.e., genes with a scaled intramodular connectivity (IMK) score of 0.9 or greater, of which 20 have no BP GO term annotation and 12 were described as “uncharacterized” or “unknown”. Module Green1 had the highest number of hub genes, 26, well above the mean number of hub genes per module, 3.18 (SD ± 3.37) (see Appendix A).

In order to relate and integrate the modules within a biological context, we performed GO term enrichment tests on all 62 modules. We observed that all modules were significantly enriched with one or more GO terms (adjusted *p*-value < 0.05), with the largest module (Green1) enriched in GO terms such as “growth”, “cell division”, and “plant-type cell wall biogenesis”, among others. The complete GO enrichment data for the 62 modules can be found in Appendix A. To test the network’s performance, we evaluated its capacity to connect genes with shared GO terms. Using the gene adjacencies matrix, we obtained a mean Guilt-By-Association-AUROC (GBA-AUC) of 0.82. Moreover, using the simple grouping by modules as a network, we still measured a robust mean GBA-AUC of 0.70. Furthermore, as a supplementary quality measure, we found a negative correlation between gene connectivity and the dN/dS ratio (Appendix A). To further characterize the network and check its consistency, we investigated the two modules with the tightest correlation between their genes (highest “density”) after correction by module size, Green20 and Green35 (Appendix A). Both modules were highly enriched for “photosynthesis”, as well as for cellular compartments such as “thylakoid”, “photosynthetic membrane”, and “chloroplast”. These results are in line with the prime biological functions of plant green tissues, thus attesting to the reliability of GreenGCN as a reference co-expression network to explore relationships among genes.

### 2.2. Construction of a Weighted Gene Co-Expression Network from Stressed and Control Root Tissue

In order to study the root tissue under controlled and stressed conditions, we generated a new network. We collected a total of 163 samples, of which 134 were “control” conditions and 29 corresponded to “biological disturbances”, namely infections with the fungal pathogen *V. dahliae* (14), arbuscular mycorrhizal fungi *Rhizoglomus irregulare* (6), and the parasitic plant *Orobanche cumana* (9) (see Appendix A). The initial cohort of 163 samples from 11 different studies was filtered down to 150 samples (Table 1), containing 20,850 genes. We selected a soft threshold β = 6 resulting in a scale-free network fit of R^2^ = 0.92. The resulting network, from now on referred to as “RootGCN”, had 77 modules (named Root1 to Root77), with the largest module (Root1) grouping 8.36% of genes and only 2.68% of genes remaining unconnected in module Root0. We observed 293 “hub-genes” with IMK ≥ 0.9, of which 82 were genes with no BP GO term annotation and 23 were described as “uncharacterized” or “unknown function”. The mean number of “hub genes” per module was 3.77 (SD ± 3.0). In this network, modules Root3 and Root24 had the highest number of hub genes, 16 and 14, respectively (see Appendix A). In addition to identifying modules enriched in biotic defense functions, we anticipated discovering co-expression modules that display correlations specific to the comparison between control versus infected samples. GO term enrichment tests showed that all modules were enriched significantly with one or more GO terms (*p*-value < 0.05) (Appendix A). Interestingly, we found module Root5 highly enriched for root development and root hair elongation, whereas, as expected, we observed no enrichment for chlorophyll biosynthesis in any module. Applying the same Guilt-by-Association prediction method as before, we measured a GBA-AUC of 0.79 using the adjacency matrix. Moreover, we again observed an inverse correlation between gene connectivity and dN/dS ratios, as expected (Appendix A). To further check consistency in this network, we again analyzed the highest density module, Root37, which is highly enriched in cell wall biosynthesis processes, an important process in root development [28] (Appendix A).

### 2.3. Identification of Modules Related to Fungal Resistance

Since we were interested in defense responses against pathogens, particularly against phytopathogenic fungi, we selected modules enriched in immune functions, and as reinforcement, we also searched for enrichment in WRKY genes, given they have been frequently linked to defense responses [29]. We first searched for modules significantly enriched in the GO term “response to fungus” (GO:0009620) and its direct descendants. We identified four modules: Green16, Green18, Green24, and Root18, specifically enriched in the term “defense response to fungus”, among others. Expanding our scope, we searched the term “defense response” (GO:0006952) and all its child terms, also detecting modules Green5, Green39, and Root44. Modules Green5, Green18, Green24, Green39, Root18, Root36, and Root44 were all significantly enriched for WRKY-like genes, and furthermore, these modules’ eigengenes were all positively correlated among each other (within each GCN) (Figure 2; see also Appendix A for full eigengene correlations in GreenGCN and RootGCN, respectively). Interestingly, we observed that module Green16 was inversely correlated to Green5, Green18, and Green39 while harboring no WRKY-like proteins.

Furthermore, in the RootGCN, it was possible to test trait-to-module correlation using the sample’s condition (control or infected) as a trait. Seven modules were significantly correlated (adjusted *p* value < 0.01), in descending order: Root0, Root36, Root21, Root1, Root14, and Root35. None of these modules were significantly enriched for defense-related functions, but it caught our attention how the module Root36 was enriched in WRKY-like genes (Table 2) and how its eigengene was inversely correlated to modules Root18 and Root44 (Figure 2).

Among the total 119 WRKY-like genes in the annotation of the sunflower genome, 37 were present in GreenGCN and 66 in RootGCN, of which 22 and 12 were included in the previously mentioned modules (Green5, Green18, Green24, Green39, Root18, Root44, and Root36). WRKY-like genes were not only enriched in these modules; however, some also had central roles in them, as evidenced by their high IMK (Table 2). For example, module Green24 was the most enriched in WRKYs, harboring 8 out of 185 total genes, with four of them having an IMK > 0.6. Inherently, most of the sunflower homologs to *A. thaliana* WRKYs in the modules of interest were described as being involved in defense responses against pathogens (Table 2). As another layer of evidence, only nine of the hub genes (IMK > 0.8) inside these modules were previously annotated as related to defense functions; yet, we identified 16 hub genes, so far not annotated as defense-related, that had homologs in *A. thaliana* linked to such processes (Table 3; also see Appendix A).

### 2.4. Module Preservation among GreenGCN and RootGCN

We noticed how modules of interest from the RootGCN were enriched in genes also present in modules of interest from the GreenGCN (Table 4); this again suggested the possibility of modules being conserved between the two tissues. To test this hypothesis, we analyzed the topology preservation between the Green and Root GCNs.

There were 14,481 genes present in both networks, with overall similar expression and connectivity ranks between networks (see Appendix A) and significant Pearson correlation coefficients of 0.68 and 0.5, respectively. Using GreenGCN modules as the reference, 31 out of 63 seem markedly preserved (Zsummary score > 10 and medianRank < 40), and two appear markedly not preserved (Zsummary score < 2). On the one hand, module Green14 (involved in cellular metabolic compound salvage in mitochondria) is well preserved (Zsummary score of 69.4 and medianRank of 3), as may be expected. On the other hand, Green47, with a Zsummary score of 1.2 and 40 genes, is involved in cellular biosynthetic processes in the chloroplast envelope and is markedly not preserved. Moreover, another module involved in photosynthesis, Green20, has a Zsummary score of 4.3 with 94 genes and is not preserved as we would expect.

Among the modules of interest we identified, the Green24 and Root44 modules were particularly preserved, having Zsummary scores > 10 and relatively low medianRank scores (Table 5). On the contrary, the modules Green5, Green18, and Root18 didn’t pass the Zsummary threshold despite having larger module sizes, as also reflected by their medianRank. Modules Green16, Green39, and Root36 showed marked non-preservation. This suggests that some possible defense-related processes influenced by WRKY genes (modules Green24 and Root44) have similar co-expression patterns in both studied tissues.

### 2.5. Functional Prediction of “Unknown/Uncharacterized” Genes in Defense Modules

The selected eight defense-related modules (Green5, Green16, Green18, Green24, Green39, Root18, Root36, and Root44) included a total of 431 genes with no GO term annotation and 162 genes coding for proteins described as having “unknown” or “uncharacterized” functions (HeliaGene Sunflower annotation). Interestingly, by simply searching for the closest homologs in model species, we found that 125 of these unannotated genes had homologs associated with defense functions (Appendix A). Moreover, 30 genes from this subgroup had an IMK > 0.6, meaning they may play important roles in these gene clusters. These results reaffirmed our interest in the selected modules and show the value of constructing co-expression networks to characterize gene functions, particularly for under-annotated biological processes such as “defense response”.

Therefore, we used our co-expression data to perform functional inference to further characterize *loci* of interest. We used a strategy that combined two co-expression-based functional annotation algorithms (EGAD and vHRR; see Materials and Methods). This approach rendered 881 genes predicted by both methods with defense-related GO terms, of which 482 were included in at least one module of interest (Appendix A). More interestingly, 65 of the 881 newly annotated genes were previously described as having “unknown function”, with 10 of them being central genes (IMK > 0.7). From the 119 WRKY-like genes annotated in the sunflower genome, 68 were included in at least one of our GCNs, of which 16 were predicted by both methods to be related to defense responses. Of these proteins, 15 were present in modules of interest (see Appendix A).

### 2.6. Functional Prediction of Candidate Genes Associated with Resistance to V. dahliae and S. sclerotiorum

After completion and analysis of our networks, we compiled lists of candidate genes previously associated with resistance to the sunflower fungal pathogens *V. dahliae* and *S. sclerotiorum* to assess the performance of our networks in predicting gene functions. We collected three groups of genes: Group A, a list of 26 candidate genes from Filippi et al. [25]; Group B, genes that were close (±1 megabase) to markers from four QTL mapping and genome-wide association studies: Zubrzycki et al. [30], Filippi et al. [25], Filippi et al. [31], and unpublished data from Montecchia et al.; and Group C, genes that were in between markers flanking QTL regions from the work of Talukder et al. [32]. It is noteworthy that groups B and C employed different germplasm in their respective studies, which allows for a broader range of genetic characteristics to be captured. The full list of genes totaled 12,485 unique candidate *loci*, of which 5101 genes were present in at least one of the studied GCNs, with 439 genes being included in our predefined modules of interest. Finally, at this intersection, we focused only on the genes that were also predicted to be related to defense functions by EGAD and vHRR methods. The resulting list included 122 unique genes, of which 1 was from group A, 40 were from the close vicinity of 43 associated markers from group B, and the remaining 81 genes were included in 13 QTL regions from group C (see Appendix A). Interestingly, 7 of the 122 genes are linked to markers from different studies, giving further support to their biological significance. Furthermore, of the 122 total *loci*, 43 had no previous BP GO term annotation, 3 were WRKY transcription factors, and 8 were described as “unknown” or “uncharacterized” proteins.

## 3. Discussion

To our knowledge, this is the first large-scale multi-study co-expression network reported for sunflower. The objective was to arrange genes into biologically significant clusters while investigating their connections, classify previously reported candidate *loci* and deduce their possible function, and discover new potential candidates for fungal disease resistance. For this, we constructed two weighted gene co-expression networks (GCNs) from different tissues of sunflower plants: green healthy tissue and stressed/healthy root tissue. It has been shown that merging samples from different tissues and studies generates networks with less gene connectivity and fewer modules [33]. Hence, in our work, we decided to separate samples according to tissue origin and construct different networks to compare them later. Moreover, it is known that in both simulated and real datasets, approximately 100 samples or more are required to capture transcriptional genome-wide regulations in a consistent and robust manner [34]. Therefore, we did not undertake the analysis of other relevant tissues, such as the stem of fully grown plants or capitula, since there were not enough samples to construct robust networks.

Co-expression network analyses have become a popular tool to find associations between complex expression data in plants [35,36,37,38,39,40]. The use of public transcriptomic data, while increasing the heterogeneity, also increases sensibly the sample size, enhancing the possibility of establishing genuine associations and providing valuable insights into gene co-expression patterns and their potential roles in defense responses in sunflower. In our work, the inclusion of 733 samples in the GreenGCN made it possible to organize 25% of the genes in the sunflower’s genome into 62 modules, while the inclusion of 163 samples in the RootGCN grouped 40% of the genes in the sunflower’s genome into 77 modules. Both networks exhibited a scale-free topology, indicating a robust organization of gene co-expression [41]. The identification of hub genes with high intramodular connectivity scores suggests their importance in coordinating gene expression within the modules. To further establish a biological context for the modules, a GO term enrichment analysis was performed. All modules of both networks showed significant enrichment for one or more GO terms, further supporting the functional relevance of the identified gene clusters. Notably, module Green1 stood out with the highest number of hub genes, indicating its potential significance in the biological processes of growth, cell division, and plant-type cell wall biogenesis. Moreover, the GO term analysis of the RootGCN modules revealed significant enrichment for defense responses, consistent with the control versus infected condition of the samples. Additionally, the networks’ performances were evaluated by measuring GBA-AUC, which quantifies each network’s ability to connect genes with shared GO terms. The obtained GBA-AUC values indicated a high level of functional coherence within each network. The analysis revealed a negative correlation between gene connectivity and dN/dS ratios, consistent with Masalia et al. [42], where genes with less overall connectivity appeared to be under more intense selective pressure. In summary, the networks serve as a valuable tool for investigating potential connections between genes and advancing towards the field of systems biology.

The identification of modules related to fungal disease resistance was a key focus of the study. We identified eight defense-related modules significantly enriched in GO terms associated with defense responses and/or WRKY genes: Green5, Green16, Green18, Green24, Green39, Root18, Root36, and Root44. In general, these modules were represented by genes with assorted functional descriptions. They included signaling proteins (e.g., receptors, kinases, and transporters), ATPases, heat shock proteins, transcription factors, and carbon metabolism proteins, among many others. Additionally, some of these genes are characterized by their domains. As such, 37 putative leucine-rich repeat proteins, which are involved in different types of interaction and are often related to immune responses, were represented in all modules but one, Green16 [43]. Besides, 24 *loci* involved a calcium-binding domain, required for calcium sensing in early events of plant defense responses [44]. Both are consistent with effector-triggered immunity (ETI) and PAMP-triggered immunity (PTI). Pathogenesis-related (PR) proteins have a pivotal role in building the stress response of plants [45]. However, only two genes are annotated as such in the sunflower’s genome, and none were present in our networks. Nonetheless, these modules contain genes described as chitinases or glucanases, suggesting the presence of unannotated PR proteins among these modules. The intramodular connectivity scores within the modules pointed out the main genetic drivers of each as hub genes. Among the 40 hub genes with predicted function, nine were directly associated with defense responses through their GO-Term annotation, while the other 16 hub genes were linked to them through their *A. thaliana* homologs. An example worth mentioning is the presence of a probable EDS1L (Enhanced Disease Susceptibility 1) protein as a positive regulator of basal resistance and mediated by Toll-interleukin-1 receptor-nucleotide binding-leucine-rich repeat (TIR-NB-LRR) resistance proteins in Green5 [46,47], in conjunction with other signaling proteins. Furthermore, transcription factors representing different families had a central position, e.g., NAC, bHLH, AP2/B3-like, and WRKY, with the latter particularly represented. Additionally, the positive correlation between the eigengenes of these modules within each GCN suggested a coordinated regulation of defense processes. Notably, module Green16 exhibited an inverse correlation with other defense-related modules, indicating potential functional distinctions. What is more, no WRKY genes were found in this module. The WRKY family is considered one of the largest transcription factor (TF) families in higher plants [29,48], with 119 *loci* in sunflower [7,49]. Members of the WRKY family are involved in several stress-related processes, including defense responses and biotic or abiotic stress responses [48]. However, only eight sunflower WRKY genes are annotated with the GO term “defense response” or its child terms in the HanXRQr1.2 genome. In sunflower, previous findings by Giacomelli et al. [50], Liu et al. [49], Li et al. [51], and Filippi et al. [52] implicated WRKY genes in fungal defense responses. In this study, we also identified homologs of sunflower WRKY genes in our networks that were previously associated with defense responses in other plant species (see Table 2). The presence of these sunflower gene homologs within the “defense modules”, along with their high intramodular connectivity, provides evidence for their function in a defense regulatory role. In support of the predictive ability of the networks, the WRKY hub of the Green24 module, HanXRQChr16g0509771 (IMK = 0.89), without prior GO defense-related annotation, was found by Peluffo et al. and Giacomelli et al. (under the name HaWRKY7) to be differentially expressed upon infection with *S. sclerotiorum* in two independent studies [50,53]. Furthermore, using this candidate gene as a marker in an association mapping study, Filippi et al. (2015) found it to be associated with low severity of Sclerotinia head rot [52]. Moreover, another five WRKY genes (four present in Green24 and one in Green39) were described by Giacomelli et al. [50] to have differential expression under *S. sclerotiorum* and *Pseudomonas syringae* infection: HanXRQChr16g0499381, HanXRQChr03g0088861, HanXRQChr03g0084521, HanXRQChr08g0216831, and HanXRQChr16g0505941 (IMKs = 0.64, 0.37, 0.37, and 0.18, respectively) (Appendix A). Lastly, the WRKY-like genes HanXRQChr14g0460611, HanXRQChr15g0480431 and HanXRQChr11g0348481 from modules Green18/Root44, Green5/Root18 and Root18, respectively, have been shown to be upregulated in roots under *R. irregulare* infection [49]. It is worth mentioning that among the total of 26 WRKY genes identified in the studied modules, eight genes are found in both the Green and Root modules of interest. This finding suggests a potential link or association between these gene clusters in different tissues. Our comprehensive analysis, based on public data, presents compelling evidence supporting the involvement of multiple sunflower WRKY genes in defense responses.

To illustrate the functional potential of defense-related networks, we selected the gene AT3G55800, a homolog of the hub gene HanXRQChr14g0459921 in the Green16 module, which is described as a sedoheptulose biphosphatase. According to the Protein-Protein Interaction Networks of the STRING Database (© STRING CONSORTIUM 2023), this protein has been reported to be coexpressed with several fructose-bisphosphate aldolases and glyceraldehyde-3-phosphate dehydrogenases, among others. In addition to its role in the process of carbohydrate biosynthesis, this protein, along with three other metabolic enzymes involved in glycolysis and the pentose phosphate pathway, decreased during the response to *Pseudomonas syringae* pv tomato DC3000 in *A. thaliana* leaves [54]. At the same time, the ATP synthase subunit CF1δ, which is encoded by the gene AT4G09650 and plays a central role in oxidative or photosynthetic phosphorylation, decreased in abundance. Remarkably, the Green16 module is highly enriched in predicted enzymes of carbohydrate metabolism, including several fructose biphosphatases, glyceraldehyde-3-phosphate dehydrogenases, sedoheptulose biphosphatases, and malate dehydrogenases, many of which play a central role in this module. Module Green16 also contains a homolog of AT4G09650, HanXRQChr01g0001181. Moreover, this gene was down-regulated in a tolerant inbred line at early stages of infection with the fungus *S. sclerotiorum* in sunflower, suggesting a common functional role in pathogen-associated molecular pattern (PAMP)-triggered immunity [55]. Several authors have identified modulation of primary metabolic pathways during the defense response, although the specific players in each species remain to be elucidated [56,57,58].

Previous studies identified various *loci* involved in the quantitative resistance to *V. dahliae* and *S. sclerotiorum*, mostly through comprehensive analysis in biparental mapping, association mapping, or transcriptome differential expression studies [25,26,30,31,32,55]. In an attempt to provide a biological context to the candidate genes derived either by co-localizing with a molecular marker in GWAS studies or by being located in between markers in QTL studies, we focused on the candidates predicted to be related to defense functions in our modules of interest. This strategy allowed us to select and annotate by association the candidates immersed in a defense-related biological context. The resulting list included three newly annotated WRKY transcription factors along with eight previously “unknown” or “uncharacterized” proteins with new defense-related GO terms. One interesting result is HanXRQChr16g0524031, which is a candidate gene from group B with an original description of “Protein of unknown function (DUF3133)” and now has the new GO term “regulation of defense response”. To further examine this locus, we looked at its closest homologue in *A. thaliana*, AT3G61670 (e-value < 6 × 10^−124^), which also has the DUF3133 domain and is described as an extra-large G-like protein. Interestingly, extra-large G-like proteins are known to be involved in stress response [59], and moreover, this homologue has the GO term “regulation of defense response to fungus” among its annotations. In addition, we investigated another candidate gene from group C, HanXRQChr05g0150771, which is an uncharacterized protein conserved in plant genomes and supported by expression data. The *A. thaliana* homologue AT5G42860 (e-value < 3 × 10^−100^) is a “Cysteine/Histidine-rich C1 domain family protein” with strong expression in roots (BAR eFP Browser, The Arabidopsis Information Resource (TAIR)) and has GO annotations of “response to jasmonic acid”, “response to wounding”, and “root development”.

Overall, the aforementioned findings present a novel approach to exploiting the great amount of data generated in sunflower to date. Although only a moderate proportion of the sunflower genome could be included in our networks, they highlight the presence of genetic drivers and group co-expressed elements, expanding our understanding of the associations established during different biological processes taking place. The insights into disease resistance in sunflowers are of particular interest for breeding programs. The identification of WRKY genes with a recurrent and central role in defense-related modules presents a compelling example of promising breeding targets. The dominant role of these transcription factors can potentially achieve more comprehensive and substantial effects on multiple *loci* associated with defense responses by modulating the expression of various downstream genes. This could also apply to other hub-genes, allowing for fine-tuning the breeding strategies and leading to the development of sunflower varieties with enhanced defense capabilities against fungal pathogens. The defense-related annotation resource generated in this study is available in Appendix A, where it can be explored by the sunflower and plant communities to facilitate future research, as these novel annotations might provide the missing link in defense-related pathways.

## 4. Material and Methods

### 4.1. Data Acquisition

We explored all the publicly available sunflower RNA-seq samples from the Sequence Read Archive (SRA) (NCBI) and downloaded samples originating from either: (1) photosynthetic tissues under control conditions, and (2) radical tissues under either control conditions or infected by pathogens. The total of 898 samples comprised 31 SRA studies and 391 different genotypes, detailed in Appendix A. SSR markers from Zubrzycki et al. [30] were mapped to the sunflower reference genome HanXRQr1.2 [7] to identify their genomic position.

### 4.2. Quality Control and Mapping of Data

The sequencing quality of each sample was evaluated with FastQC [60], and any adaptors and low-quality reads were removed with Trimmomatic [61]. We quantified the expression of our reference transcriptome via Salmon v1.5.2 [62], using the genome HanXRQ version r1.2 as our reference to produce one unique transcript per gene (coding and non-coding), totaling 56,917 distinct sequences [7]. The approach used by Salmon estimates abundance uncertainty due to random sampling and the ambiguity introduced by multi-mapping reads, a key feature when mapping to a transcriptome reference. The mapping rates ranged from 1% to 97%, with a median of 75%. Overall, samples from wild-cultivar species had lower mapping percentages, than samples from infected tissues, probably due to the low efficiency of RNA extraction from sunflowers. Samples with less than 3 million read counts were discarded.

### 4.3. Co-Expression Network Analysis

For each group of samples, we applied a recursive process of: filtering genes with less than two counts per million (CPM) in ¾ of the samples; normalizing read counts via variance stabilizing transformation from the “DESeq2” R package [63]; adjusting batch effects arising from sample origin via an empirical Bayesian method described by Johnson et al. [64] implemented in the R package “sva” [65]; and excluding visual outliers in hierarchical clustering of Pearson distances. With these matrices, we calculated weighted gene co-expression networks using the R package “WGCNA” [27]. We used bicor correlation, signed hybrid weighting, and signed Topological Overlap Matrices. Appropriate β soft thresholds were picked by maximizing the networks’ fit to a free scale topology. Merge cut height and deep split parameters were 0.25 and 2 respectively, and minimal module size was set to 30.

### 4.4. Guilt-by-Association Network Performance Evaluation

We used the EGAD R package [66] to evaluate the network’s capacity to connect genes with shared GO terms as a measure of quality. Based on the Guilt-by-Association principle, networks can be evaluated by hiding a subset of genes’ GO terms and testing whether the hidden GO terms could be predicted from the remaining annotated genes given their expression correlation. GO data for the HaXRQr1.2 sunflower genome was downloaded from https://www.heliagene.org/ (accessed on 3 March 2022). For evaluation, only GO terms with 20 to 300 annotated genes present in the evaluated network were used, amounting to between 556 and 624 GO terms, depending on the network. The predictions’ performances were measured by AUROC values in 5-fold cross-validation.

### 4.5. Estimation of dN/dS Ratios

We estimated the molecular evolution rates of each sunflower gene using PAML’s yn00 model [67] with *A. thaliana* as a reference. We downloaded *A. thaliana* CDS from Phytozome v12 (https://phytozome-next.jgi.doe.gov/ (accessed on 3 March 2022)), translated the sequences, and identified putative sunflower orthologs by using reciprocal best BLASTP hits with an e-value threshold of 1 × 10^−10^. Then, the resulting 10,425 orthologous pairs were aligned using MUSCLE v.5.0.1428 [68]; we used pal2nal [69] to transform the protein alignments into nucleotide alignments before feeding them to the PAML algorithm. Rates of non-synonymous substitutions per non-synonymous site (dN), synonymous substitutions per synonymous site (dS), and estimates of adaptive evolution (ω = dN/dS) were compared via linear regression against estimates of gene connectivity in each network.

### 4.6. GO Enrichment Analysis

GO enrichment of modules was analyzed via the R/Bioconductor package TopGO [70] using Fisher’s exact test as a statistic (considering the complete list of 56,917 unique genes as background references) and two methods of dealing with the GO graph structure: “classic” and “weight”. *p*-values below 0.05 were considered significant enrichments; *p*-values calculated with the “weight” method do not require multiple testing correction, as per the authors’ indications [71].

### 4.7. Module-Condition Relationship

To identify modules that were significantly associated with biological conditions, correlations between module eigengenes (MEs) (i.e., the first principal component of the module, which represents the overall expression level of the module) [72] and conditions were computed using binomial generalized linear models implemented in R in the base package stats.

### 4.8. Module Preservation Analysis

To assess the preservation of modules from one co-expression network to another, we used the WGCNA package function modulePreservation to calculate the composite preservation statistics “Zsummary” and “medianRank”. Higher values of Zsummary indicate stronger preservation of the module being compared between the networks. Zsummary scores higher than 10 indicate consistent preservation of the module, while scores lower than two indicate consistent disruption. However, Zsummary tends to be dependent on module size, meaning bigger modules will have inflated Zsummary scores since it is more significant to observe preserved connectivity patterns among hundreds of genes than to observe the same among only a few [73]. On the other hand, medianRank is size-independent, and lower values mean more preserved modules; however, medianRank is dependent on the total number of modules. As a reference, modules with medianRanks higher than ⅔ of the total module number are considered non-preserved [74].

### 4.9. Gene Function Prediction

We used the mentioned EGAD package along with the methods of vHRR [13] to predict the biological functions of network genes.

For the EGAD method, we used the same adjacency matrices calculated by WGCNA. For each network, the optimum threshold for each GO term was picked according to the maximum F1 score, calculated using the known annotations. For the vHRR pipeline, we used the filtered and normalized data that was also used as input for WGCNA. As input annotations, we used the aforementioned GO data. The performances of the prediction methods were evaluated by F1 scores over the known annotations.

The overall performance comparison of these methods is shown in Appendix A. In order to prioritize genes with possible defense functions, we chose to consider both methods complementary. We selected a list of BP GO terms linked to defense response and related terms to predict new annotations. This list is detailed in Appendix A, page 2. GO terms not present in any of the genes from the GCNs are omitted. Our strategy was to focus on genes that were annotated by both methods with any defense-related GO term. That is, for example, we would select a gene predicted as “defense against fungus” by EGAD using the GreenGCN data and “response to stress” by vHRR using the RootGCN data.

## Figures and Tables

**Figure 1 plants-12-02767-f001:**
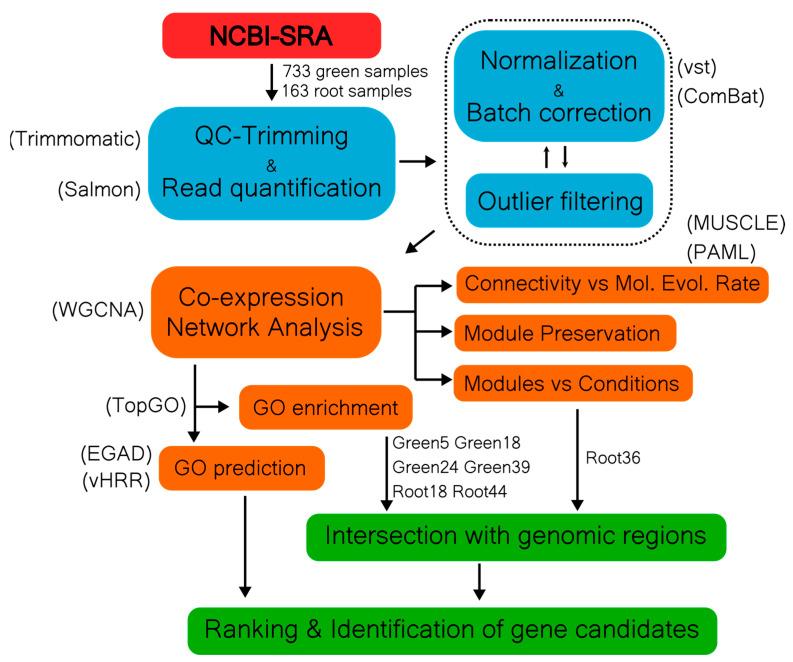
Bioinformatic flowchart describing sample processing in our co-expression network analysis.

**Figure 2 plants-12-02767-f002:**
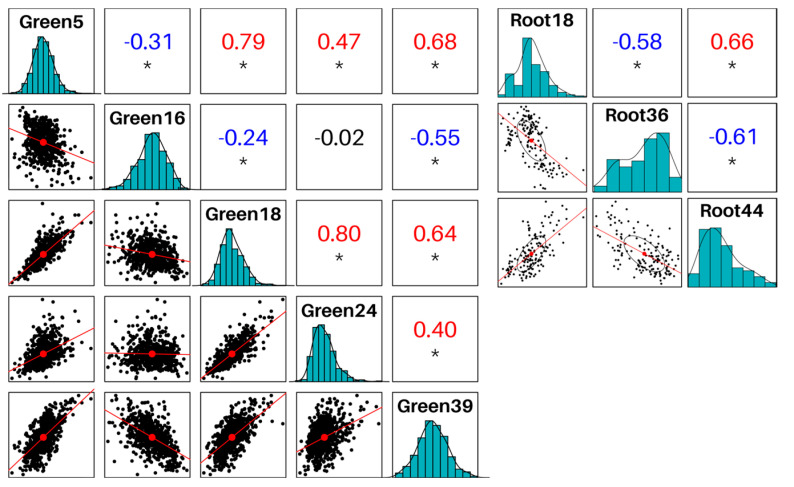
Eigengenes correlations between modules of interest in GreenGCN and RootGCN networks. Asterisks indicate a *p*-value < 1 × 10^−10^.

**Table 1 plants-12-02767-t001:** Number of samples used for the construction of the GreenGCN and the RootGCN.

	Samples Condition	Initially	Post Filtering
GreenGCN	Healthy	733	686
RootGCN	Healthy	134	127
	*V. dahliae* infection	14	14
	*R. irregulare* infection	6	6
	*O. cumana* infection	9	3

**Table 2 plants-12-02767-t002:** Putative WRKY genes present in modules of interest and their corresponding closest homologs in *A. thaliana*. Asterisked genes of *A. thaliana* are specifically associated with defense responses against pathogens, according to the TAIR database.

Sunflower Gene			Closest Homolog in *A. thaliana*			
Gene Name	Module	IMK	Homolog	Identity %	Q-Cov %	E-Value
HanXRQChr10g0294651	Green5	0.3	AtWRKY70 *	43.31	19	2 × 10^−23^
HanXRQChr16g0506841	Green5	0.28	AtWRKY6	54.02	81	8 × 10^−137^
HanXRQChr16g0508961	Green5	0.26	AtWRKY7 *	49.72	95	1 × 10^−95^
HanXRQChr08g0211091	Green5	0.21	AtWRKY4 *	53.71	97	2 × 10^−144^
HanXRQChr15g0480431	Green5/Root18	0.13/0.18	AtWRKY28 *	40.48	100	3 × 10^−61^
HanXRQChr11g0329641	Green5	0.08	AtWRKY41 *	46.55	39	1 × 10^−29^
HanXRQChr10g0306731	Green5	0.07	AtWRKY1 *	43.62	65	6 × 10^−68^
HanXRQChr10g0281391	Green5	0.05	AtWRKY21	48.11	100	2 × 10^−106^
HanXRQChr03g0071411	Green18	0.71	AtWRKY51 *	53.78	80	8 × 10^−39^
HanXRQChr14g0460611	Green18/Root44	0.41/0.78	AtWRKY70 *	42.94	54	2 × 10^−37^
HanXRQChr04g0113641	Green18	0.09	AtWRKY21	49.62	100	5 × 10^−113^
HanXRQChr16g0509771	Green24	0.89	AtWRKY33 *	46.56	88	2 × 10^−117^
HanXRQChr09g0274431	Green24/Root18	0.69/0.97	AtWRKY40 *	53.17	99	8 × 10^−99^
HanXRQChr06g0166901	Green24	0.67	AtWRKY41 *	37.25	100	3 × 10^−46^
HanXRQChr16g0499381	Green24/Root18	0.64/0.85	AtWRKY40 *	51.49	99	1 × 10^−93^
HanXRQChr03g0084521	Green24	0.37	AtWRKY11 *	42.01	91	2 × 10^−46^
HanXRQChr03g0088861	Green24	0.37	AtWRKY6	47.67	93	7 × 10^−132^
HanXRQChr05g0142161	Green24	0.28	AtWRKY11 *	37.70	95	2 × 10^−44^
HanXRQChr08g0216831	Green24/Root44	0.18/0.24	AtWRKY70 *	40.19	63	4 × 10^−37^
HanXRQChr08g0228641	Green39/Root18	0.79/0.38	AtWRKY33 *	46.65	91	3 × 10^−113^
HanXRQChr09g0264011	Green39/Root36	0.31/0.01	AtWRKY4 *	43.05	97	3 × 10^−107^
HanXRQChr16g0505941	Green39/Root18	0.29/0.05	AtWRKY7 *	47.15	94	1 × 10^−81^
HanXRQChr11g0348481	Root18	0.17	AtWRKY53 *	40.96	98	1 × 10^−74^
HanXRQChr17g0544771	Root36	0.71	AtWRKY72 *	38.29	81	6 × 10^−60^
HanXRQChr11g0336511	Root36	0.42	AtWRKY6	36.49	94	1 × 10^−72^
HanXRQChr08g0209791	Root44	0.32	AtWRKY75 *	92.13	35	2 × 10^−55^

**Table 3 plants-12-02767-t003:** Hub genes (IMK > 0.8) from modules of interest and their closest homologs in *A. thaliana* related to defense functions.

Sunflower Gene			Closest Homolog in *A. thaliana*			
Name	Module	IMK	Homolog	Identity %	Q-Cov %	E-Value
HanXRQChr01g0014161	Green5	1.0	AT5G48380	60.54	90	0
HanXRQChr04g0107431	Green5/Root44	0.96/0.9	AT1G34420	48.14	99	0
HanXRQChr02g0045301	Green5	0.82	AT3G48090	38.99	98	1 × 10^−147^
HanXRQChr09g0248321	Green24/Root18	0.94/0.60	AT5G12010	67.32	89	0
HanXRQChr02g0040711	Green24	0.89	AT1G18740	60.21	98	5 × 10^−163^
HanXRQChr15g0496321	Green24	0.89	AT2G40140	51.39	94	1 × 10^−178^
HanXRQChr03g0086901	Green24	0.88	AT3G56880	34.54	99	2 × 10^−18^
HanXRQChr16g0504131	Green24	0.86	AT2G40140	51.25	97	0
HanXRQChr13g0399921	Green39	1	AT3G09830	66.92	88	0
HanXRQChr12g0366961	Green39	0.81	AT1G30755	49.69	100	0
HanXRQChr10g0300021	Root36	0.96	AT5G01050	53.74	98	0
HanXRQChr05g0161441	Root36	0.94	AT1G22400	52.50	98	0
HanXRQChr07g0205991	Root44/Green18	1.0/0.39	AT1G08450	70.12	95	0
HanXRQChr17g0553831	Root44	0.89	AT5G42510	42.47	64	1 × 10^−38^
HanXRQChr01g0001251	Root44	0.84	AT3G54040	54.02	70	5 × 10^−58^
HanXRQChr04g0123531	Root44	0.83	AT3G60450	55.42	93	5 × 10^−93^

**Table 4 plants-12-02767-t004:** Shared genes between modules of interest. Asterisks indicate significant enrichment (*p*-value < 0.01) by the Fisher test.

Modules	Green5	Green16	Green18	Green24	Green39
Root18	20 *	1	34 *	24 *	5
Root36	6	0	1	0	3
Root44	19 *	0	12 *	1	1

**Table 5 plants-12-02767-t005:** Preservation of modules of interest between GreenGCN and RootGCN. Modules Green24 and Root44 showed signs of consistent preservation among the two GCNs.

Modules	Green5	Green16	Green18	Green24	Green39	Root18	Root36	Root44
Number of shared genes	530	186	247	181	90	209	62	42
Zsummary	8.81	3.12	9.24	12.67	2.06	8.58	1.95	12.65
Median Rank	58	61	51	39	56	61	72	10

## Data Availability

All data is in the public domain.

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
