# Peer review of "Co-Expression Networks in Sunflower: Harnessing the Power of Multi-Study Transcriptomic Public Data to Identify and Categorize Candidate Genes for Fungal Resistance"

_plants, 2023, doi:10.3390/plants12152767_

Round 1

Reviewer 1 Report

The article presents a meta-analysis of a large number of sunflower transcriptome experiments. The authors use co-expression networks to predict gene functions. The article is well written, the additional material is well presented. It seems to me that it is worth bringing more diagrams of Green and Root networks, perhaps as additional files, At least in the form of a general topological diagram of connections between clusters.

The font and spelling of the Latin names of plant and mushroom species should be checked. For example, Line 209, 214, 219: Arabidopsis, A. thaliana, etc.

Reviewer 2 Report

The manuscript describes a large-scale bioinformatics study of the sunflower gene co-expression network in healthy (leaf, root) and infected (root) plant tissue using publicly available transcriptomic data. As a result of a detailed analysis, the authors found probable associations between individual genes and the plant's response to fungal diseases. This study is interesting and relevant.

However, it should be noted that when analyzing the results, the authors paid the main attention to the WRKY gene family, while there are other well-known families of stress-sensitive genes. For example, it is curious whether, in the obtained data, there are genes for pathogenesis-related (PR) proteins (reviewed, for example, in DOI: 10.2174/1389203721999201231212736) that are closely associated with the response to pathogen attack in plants? Or genes of the ERF and bZIP transcription factor families (10.2174/1389203717666160619185308)?

Besides, since the authors have concentrated on the WRKY genes, could they indicate and discuss more detail the correspondence between the genes they found and the genes described in articles “Giacomelli et al. (2010) [24], Liu et al. (2020) [27], Li et al. (2020) [35] and Filippi et al 357 (2020) [18]” (L356-358) as associated with the response to fungal infections in sunflower?

To the question of links above I would like to point out that references indicated in L356-358 [“…previous findings by Giacomelli et al. (2010) [24], Liu et al. (2020) [27], Li et al. (2020) [35] and Filippi et al (2020) [18] …”] do not match the list of references:

[18] is Zhou, N.;…

[24] is Zubryzcki, J.; Fusari…

[27] is Langfelder, P.;…

[35] is Gupta, C.;…

I didn't find ‘Li et al. (2020)’ in the list of references at all, and suspect it is [Li J, Islam F, Huang Q, Wang J, Zhou W, Xu L, Yang C. Genome-wide characterization of WRKY gene family in Helianthus annuus L. and their expression profiles under biotic and abiotic stresses. PLoS One. 2020;15(12):e0241965. doi: 10.1371/journal.pone.0241965]. ?

In this regard, it would be necessary to check the remaining correspondences of links throughout the text.

Also, throughout the text it is necessary to write in italics the names of genera and species. See, for example, lines 15-16, 48, 214, 276, 277, 366, 371, 392, 393, 401, 406, etc.

L14: A. thaliana should be written as Arabidopsis thaliana L., as the name is used in the text for the first time.

L34: “Helianthus annuus L.“L.” is not in italics.

Reviewer 3 Report

In this study, the authors we constructed two weighted gene co-expression networks (GCNs) from different tissues of sunflower plants. All modules of both networks showed significant enrichment for one or more GO terms. Additionally, the networks’ performances were evaluated by measuring the Guilt-By-Association-AUROC. Particularly, there are some newly annotated genes were previously described as of unknown function”. The methods and results are acceptable. But there are some problems should be fixed. Some typical problems are listed below.

1.Major revisions:

(1) Why does GreenGCN only have data for the control conditions but not for the infected by pathogens?

(2) L203, Among the total 119 WRKY-like genes in the annotation of the sunflower genome, 37 were present in GreenGCN and 66 in RootGCN. There were 103 genes in total, and the remaining genes belonged to which group?

2.There are some writing mistakes:

L102. Please delete the fourth paragraph of the introduction, which has nothing to do with the full text.

L169, L324. Change “Gene Ontology (GO)” to “GO”. It's not the first time you don't have to write the full name.

L214, L219, L221, L347, L392, L482. “A. thaliana” should be italicized. Please check the full text, there is the same error.

L276, L405. “V. dahliae” should be italicized.

L277, L366, L371, L401. “S. sclerotiorum” should be italicized. Please check the full text, there is the same error.

L408. “(citas)” should be deleted.

L413-419 is completely repeated with L289-295.

L450. “r1.2” or “v1.2”?

3. Please use a three-line form for table 4 and 5.

4. Please Check full text, references in the article do not need to write the year, please delete.

Round 2

Reviewer 3 Report

Nice work of the revision.